# Holter ECG for Syncope Evaluation in the Internal Medicine Department—Choosing the Right Patients

**DOI:** 10.3390/jcm11164781

**Published:** 2022-08-16

**Authors:** Ophir Freund, Inbar Caspi, Yacov Shacham, Shir Frydman, Roni Biran, Hytham Abu Katash, Lior Zornitzki, Gil Bornstein

**Affiliations:** 1Internal Medicine B, Tel-Aviv Sourasky Medical Center and Sackler Faculty of Medicine, Tel Aviv University, Tel Aviv 64239, Israel; 2Department of Cardiology, Tel-Aviv Sourasky Medical Center and Sackler Faculty of Medicine, Tel Aviv University, Tel Aviv 64239, Israel

**Keywords:** Holter, ECG monitoring, syncope, arrhythmia, internal medicine, hospital

## Abstract

Physicians use Holter electrocardiography (ECG) monitoring to evaluate some patients with syncope in the internal medicine department. We questioned whether Holter ECG should be used in the presented setting. Included were all consecutive patients admitted with syncope to one of our nine internal medicine departments who had completed a 24 h Holter ECG between 2018 and 2021. A diagnostic Holter was defined as one which altered the patient’s treatment and met ESC/ACC/AHA diagnostic criteria. A total of 478 Holter tests were performed for syncope evaluation during admission to an internal medicine department in the study period. Of them, 25 patients (5.2%) had a diagnostic Holter finding. Sinus node dysfunction was the most frequent diagnostic recording (13 patients, 52%). In multivariant analysis, predictors for diagnostic Holter were older age (OR 1.35, 95% CI 1.08–1.68), heart failure with preserved ejection fraction (OR 4.1, 95% CI 1.43–11.72), and shorter duration to Holter initiation (OR 0.73, 95% CI 0.56–0.96). There was a positive correlation between time from admission to Holter and hospital stay, r(479) = 0.342, *p* < 0.001. Our results suggest that completing a 24 h Holter monitoring during admission to the internal medicine department should be restricted to patients with a high pre-test probability to avoid overuse and possible harm.

## 1. Introduction

Evaluation of a syncopal event is challenging with a spectrum of possible etiologies; among them is cardiac-related syncope [1]. Arrhythmias are a leading cause of cardiac-related syncope, detected by an electrocardiogram (ECG) recording, preferably during a syncopal event [2,3]. Syncope guidelines recommend inpatient ECG monitoring via telemetry for selected patients with high suspicion of cardiac-related syncope [4,5]. This recommendation is based on studies with patients from the telemetry, intermediate, or syncope units. The reality in the internal medicine department, where many syncope patients are being admitted, is unfortunately different. The need for telemetry beds for a wide variety of indications (e.g., acute coronary syndrome, complicated infectious events, and respiratory deteriorations) and the lack of specialized teams to continuously watch and interpret the monitor’s recordings prompted physicians to use 24 h Holter ECG monitoring instead [6]. The yield of a 24 h Holter ECG monitoring in syncope evaluation was studied almost entirely in the ambulatory or emergency department settings, with positive findings ranging from 16% to as low as 1% [2,7,8,9,10]. It is essential to understand the diagnostic value of Holter monitoring in the internal medicine department, as Holter can lead to an extended hospital stay and higher costs [11]. The purpose of this analysis is to determine the diagnostic value of 24 h Holter ECG monitoring for the evaluation of syncope in the internal medicine department.

## 2. Materials and Methods

### 2.1. Study Design and Participants

We conducted a retrospective cohort study at a tertiary university-affiliated municipal medical center. All consecutive patients who were admitted to any of the nine internal medicine departments of the Tel Aviv Medical Center (TASMC) between 1 June 2018 and 1 June 2021 and had completed 24 h Holter monitoring during hospitalization were eligible for study participation. The inclusion criteria were: patients with syncope as the indication for Holter ECG; completion of Holter ECG during hospitalization; availability of all Holter recordings and final interpretation; and more than six hours of monitor recording (Figure 1). The study was approved by the Tel-Aviv Sourasky Medical Center review board (0876-20-TLV) and conducted in accordance with the Declaration of Helsinki. Considering the observatory nature of the study, informed consent was waived by the Tel-Aviv Sourasky Medical Center institutional review board.

For each patient, we reviewed previous medical history, reason of admission, hospitalization progress, and discharge summary. We chose to evaluate patient characteristics and clinical factors based on the previous literature [12,13]. We analyzed the syncopal episode setting (whether it occurred during activity, rest, or after a change in position) and the presence of symptoms before the episode, such as pre-syncope-related symptoms (lightheadedness, nausea, feeling hot or cold, change in vision), chest pain or palpitations. Traumatic syncope is an event with a traumatic head injury and evidence of hemorrhage on head computerized tomography. The final etiology for the syncope was determined by the discharging physician and was reviewed by our team.

### 2.2. ECG and Holter ECG

ECG recordings performed between hospital arrival and the initiation of Holter ECG monitoring were analyzed by one of the authors and compared with the interpretation of the treating physician at admission. ECG findings considered abnormal for our analysis were chosen based on previous studies [9,13,14], and they included: atrioventricular block (AVB), bradycardia (below 50 beats per minute), supraventricular tachycardia (SVT), bundle branch block (BBB), long QT interval (corrected QT interval above 460 ms), or signs of left ventricular hypertrophy (LVH). 

Twenty-four-hour Holter ECG studies were performed using monitoring devices with at least two channels (NR 302, NORAV medical, Mainz-Kastel, Wiesbaden, Germany). All Holter results had been initially interpreted during hospitalization by a cardiologist who specialized in electrophysiology and later reviewed by one of the authors. The test was considered diagnostic if: 1. It recorded any of the diagnostic findings from the list presented in Table 1, in accordance with the European Society of Cardiology (ESC) and the American College of Cardiology and American Heart Association (ACC/AHA) guidelines [4,5]. 2. The diagnostic findings did not appear in baseline ECG. 3. Holter findings affected the treatment plan. Two physicians from the research team (OF and IC) independently reviewed each case with a diagnostic Holter to ensure an effect on treatment. In case of a disagreement, the opinion of a third reviewer (GB) was used for the final decision. The Holter was considered to affect the treatment plan if, following the results, a new treatment was offered to the patient, or a chronic treatment was stopped. The decision to refer a patient for a pacemaker or an implantable cardioverter-defibrillator implantation was made after consulting with the electrophysiologist. The medical team in the internal medicine department decided on medication changes after consulting a cardiologist on a per case basis.

### 2.3. Outcomes

The primary outcome of our study was the output of diagnostic 24 h Holter monitoring. We performed a descriptive analysis of the overall results of diagnostic Holter monitoring and evaluated risk factors for the primary outcome using univariate and multivariate analyses. Secondary outcomes included duration from syncopal event or admission to Holter monitoring and the influence of the findings of Holter monitoring on treatment. 

### 2.4. Statistical Analysis

Data were analyzed with IBM SPSS statistics software version 27.0. The significance levels were set at 0.05. Data were presented as mean and standard deviation for continuous variables and as frequency and percentage for categorical variables. Chi-square tests and independent t-tests were performed to compare categorical and continuous variables between patients with and without a diagnostic Holter, respectively. Multivariate analysis for independent risk factors for diagnostic Holter was performed by logistic regression models using the backward stepwise regression, and odds ratios (ORs) with 95% confidence intervals (CIs) were calculated. The analyses included independent variables/covariates that were statistically significant in the univariate analyses. A second multivariate analysis (using the enter method) was made, with additional clinically relevant factors decided by the research team based on clinical grounds. The level of significance was set at 0.05 and was two-tailed. The goodness of fit of the model to the observed events rates was evaluated by the Hosmer–Lemeshow statistic. The relationships between hospital stay duration and other characteristics were assessed by the Pearson correlation and by a linear regression model.

## 3. Results

### 3.1. Subject Characteristics

Within the study period, 1058 24 h Holter ECG monitoring tests were performed on patients during admission to one of the internal medicine departments in our center, of whom 478 were referred for syncope evaluation and comprised the study population (Figure 1). Their baseline and clinical characteristics are presented in Table 2 and Table 3, respectively. The study population’s mean age was 75 ± 14, 55% were men, and hypertension was the most prevalent comorbidity (62%). Abnormal cardiac physical exam (auscultation) was noted in 114 patients (24%) and an abnormal ECG in 245 patients (51%). Almost one-third of the patients had the syncopal event while at rest (150 patients, 31%), and pre-syncope symptoms occurred in 226 patients (47%). Seventy-two patients (15%) were previously hospitalized after a syncopal event, and eighty-two (17%) had more than one syncopal event in the week prior to admission.

### 3.2. Holter Outcomes

Twenty-five patients (5.2%, CI 0.03–0.07) had diagnostic 24 h Holter ECG monitoring. Figure 2 presents the Holter diagnostic findings. Pauses of longer than 3 s (11 patients, 44%) were the most common Holter diagnostic findings. One patient with a diagnostic Holter finding (4%) had a correlation between symptoms (additional syncopal event) and the Holter recordings. All patients with diagnostic Holter monitoring had a change in treatment plan. Treatments following a diagnostic Holter included initiation of beta-blockers (5 patients, 20%) and a referral for a pacemaker or an implantable cardioverter-defibrillator (20 patients, 80%), of which 18 underwent implantation during hospitalization. Mean hospital stay duration and time from admission to Holter were 7.3 ± 6.4 and 2.6 ± 2.6 days, respectively (Table 2). 

We found a significant correlation between longer hospital stay and a longer time from admission to Holter, r(479) = 0.342, *p* < 0.001. A similar correlation was also found to be significant in the multivariate linear regression analysis (Table 4).

### 3.3. Predictors for a Diagnostic Holter

Older age (*p* < 0.001), atrial fibrillation (*p* = 0.004), heart failure (HF, *p* = 0.004), and a chronic use of beta-blockers (*p* = 0.004) were found to correlate best with a diagnostic Holter test (Table 2). Shorter time from syncopal event to Holter (2.04 ± 2.05 vs. 3.28 ± 3.35) and from admission to Holter (1.24 ± 0.92 vs. 2.71 ± 2.67) were also associated with a diagnostic Holter (*p* < 0.001 for both, Table 3). A multivariate regression analysis (Table 5) revealed that HF with preserved ejection fraction (HFpEF, OR 4.1, 95% CI 1.43–11.72, *p* = 0.008), older age by 5 years (OR 1.35, 95% CI 1.08–1.68, *p* = 0.008) and a shorter duration from event to Holter initiation (OR 0.73, 95% CI 0.56–0.96, *p* = 0.02) were independent predictors for a diagnostic Holter monitoring. The additional multivariate analysis, based upon clinically relevant factors, did not result in additional significant independent predictors (Table 6).

## 4. Discussion

We conducted an observational study to assess the value of 24 h Holter ECG in the evaluation of arrhythmic syncope among patients admitted to an internal medicine department. All Holter ECGs were completed during hospital stays and included diagnostic findings in 5.2% of the study population. This diagnostic value is lower compared to most previous studies, showing that inpatient ECG monitoring was diagnostic in 5–16% [8,9,10,12,15,16]. Previous studies used mainly telemetry as the ECG monitoring method, which was completed in the emergency department shortly after patient arrival [10,12,16]. The short interval between the syncopal event and the ECG monitoring contrasts with the average 2.6 days waiting time in our study population. Association between early initiation of ECG monitoring and higher diagnostic yield was established in previous studies [17], and similarly in our study (*p* < 0.001). Another explanation for the higher diagnostic yield in previous studies is a selection bias of including only patients from a syncope unit or a cardiac telemetry unit [8,18]. These special units are designated to diagnose mainly arrhythmia-related syncope [6,19], possibly leading them to admit patients with a higher pre-test probability for arrhythmia. It is also important to note that, of all patients with a diagnostic 24 h Holter monitoring in our study, only one had a correlation between symptoms and ECG recording, which is the gold standard for arrhythmic-syncope diagnosis [20]. 

Syncope is one of the most common causes of admission to the internal medicine department; hence, we decided to focus only on patients hospitalized in this setting. Patients presenting with syncope to our medical center are usually admitted to one of the internal medicine departments. This is the result of a limited number of available beds in the cardiology department and the lack of a designated syncope unit. To the best of our knowledge, this is the first study that addresses the internal medicine department patient population. We also decided to include all consecutive Holter tests to represent real-life data. 

Risk factors that correlated best with a diagnostic Holter ECG were older age, heart failure, and atrial fibrillation, all consistent with previous literature [9,12,15,18,21]. Contrary to most previous studies, heart failure with preserved ejection fraction (and not reduced ejection fraction) was found to be a risk factor in multivariant regression analysis [9,15]. We did not find any characteristic of the syncopal event to be associated with a diagnostic Holter ECG, opposite to both main syncope guidelines [4,5].

Considering the results above, it seems controversial whether Holter ECG monitoring should be completed in the non-ambulatory setting. Holter ECG monitoring is a non-invasive diagnostic test without any direct side effects, which can lead to early arrhythmia diagnosis and treatment. However, it also has the potential to prolong hospital stay by its lack of availability and the need for at least 24 h of monitoring through hospitalization. Long-term hospitalizations increase the likelihood of adverse medical events such as falls and acquired infections [22]. We found an association between longer time from admission to Holter and longer hospital stay duration (*p* < 0.001). This correlation, enforced by the multivariate linear regression results (Table 4), still does not indicate a causative relationship and might be affected by other factors, such as the level of suspicion by the ordering physician of an arrhythmia-related syncope (higher suspicion might lead to an earlier initiation). Our clinical experience shows that 24 h Holter monitoring is frequently the main rate-limiting step in the evaluation of a syncope patient, possibly extending their hospital stay. In addition, although Holter monitoring is inexpensive in terms of set-up costs, it is expensive in terms of cost per diagnosis [23]. There is no evidence to date to indicate that postponing the test to the ambulatory setting has an impact on short- or long-term outcomes. 

The 2018 ESC guidelines recommend Holter ECG use as an ambulatory test and only for patients with frequent syncope episodes (≥1 per week) [5]. A study by Brignole et al. found that by following the ESC guidelines, only 3% of all patients with syncope should have an ambulatory Holter ECG, and only 5% should undergo inpatient ECG monitoring [24]. We believe that there is an overuse of ECG monitoring, and specifically Holter monitoring, among patients with syncope. Almost one-third of the patients in our study did not have any chronic cardiac disease or abnormalities in their ECG, possibly excluding the need for inpatient monitoring. Even when high risk factors for arrhythmic syncope exist, the indication in the guidelines for inpatient monitoring refers mostly to telemetry and not to Holter. Adherence to the more basic steps in syncope evaluation, such as measuring orthostatic blood pressure, is needed before ordering more advanced tests [1,4]. 

Our study has limitations. Although we included consecutive patients in a three-year period from the nine internal medicine departments in our center, the generalizability of our results should consider the single-center nature of this study. The study was also conducted at a tertiary medical care center; hence, a referral bias cannot be excluded. The syncope evaluation process in our study was not standardized according to a specific algorithm, which could affect the decision to order the Holter ECG, while it reflects a real-world setting. A causal relationship is limited by the lack of a control group in our study. Future studies should compare the diagnostic value of inpatient Holter ECG with postponing it to the ambulatory setting and any effect on short- and long-term outcomes. 

In conclusion, the diagnostic value of in-patient Holter ECG monitoring for syncope evaluation in the internal medicine department is approximately 5%. Shorter time from admission to Holter, older age, and history of heart failure with preserved ejection fraction correlated best with a diagnostic Holter. Our results emphasize the possible overuse of Holter monitoring during hospitalization and the need for future studies to evaluate its role in the setting of the internal medicine department.

## Figures and Tables

**Figure 1 jcm-11-04781-f001:**
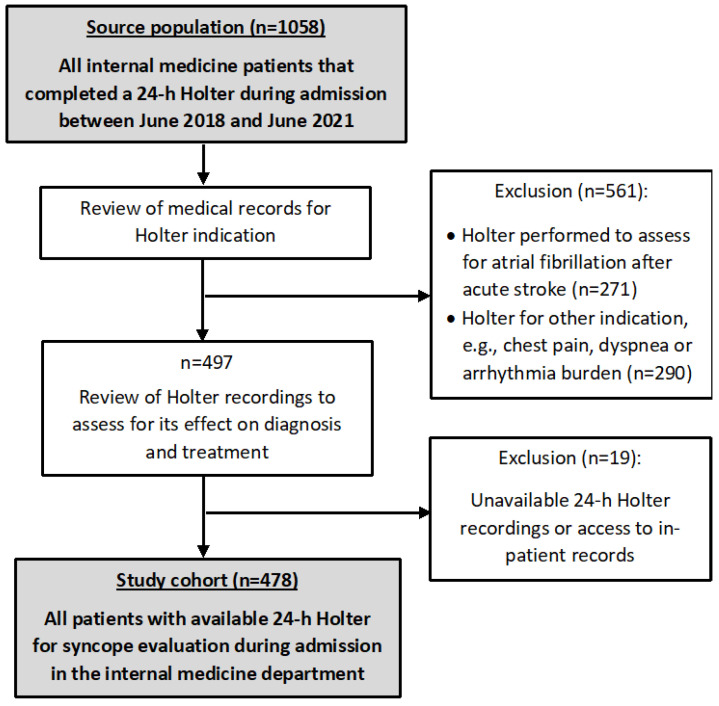
Flow chart of inclusion and exclusion process.

**Figure 2 jcm-11-04781-f002:**
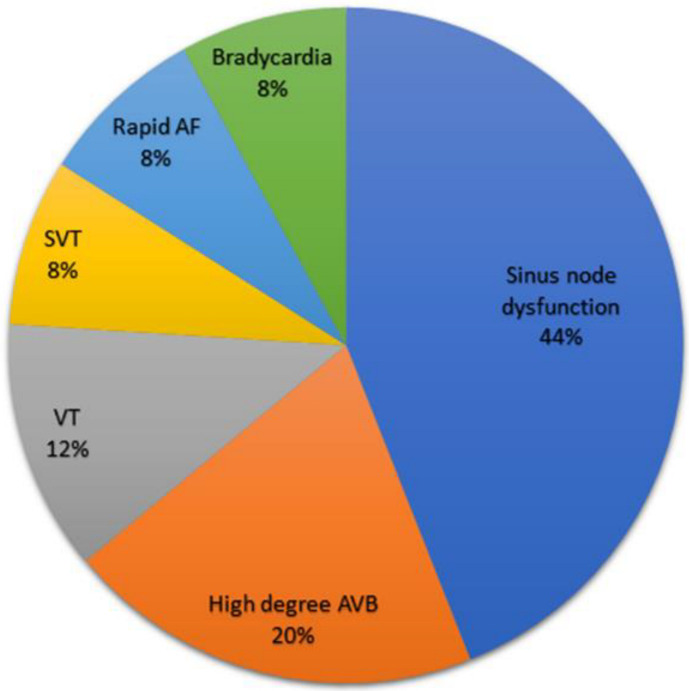
Diagnostic Holter ECG findings. Abbreviations: AVB, atrio-ventricular block; VT, ventricular tachycardia; SVT, supraventricular tachycardia; AF, atrial fibrillation. Diagnostic findings in Holter monitoring according to the main syncope guidelines, including: bradycardia under 40 beats per minute; sinus pause above 3 s; complete AV block or second-degree AV block Mobitz type 2 (high degree); and VT/SVT/AF with rate above 150 beats per minute and lasting over 32 beats. All findings were not evident in the patients’ baseline ECGs.

**Table 1 jcm-11-04781-t001:** Diagnostic Holter findings for arrhythmic syncope.

SVT or AF with a ventricular response of more than 150 bpm, lasting over 32 beats.
VT lasting over 32 beats.
Sinus pauses longer than 3 s.
Bradycardia of less than 40 bpm.
Complete AV block or second-degree AV block Mobitz type 2.
Alternating bundle branch block.

Abbreviations: SVT, supraventricular tachycardia; AF, atrial fibrillation; BPM, beats per minute; VT, ventricular tachycardia; AV, atrioventricular.

**Table 2 jcm-11-04781-t002:** Characteristics of study participants and comparison between diagnostic and non-diagnostic Holter.

Characteristic	Study Cohort *n* = 478 (%)	Diagnostic Holter *n* = 25 (%)	Non-Diagnostic *n* = 453 (%)	*p* Value
Age, mean ± SD, y	75 ± 14	82 ± 6.2	74 ± 14.5	<0.001
Female gender	217 (45)	9 (36)	16 (64)	0.332
Hypertension	295 (62)	19 (76)	276 (61)	0.131
Diabetes mellitus	140 (29)	11 (44)	129 (28.5)	0.097
Hyperlipidemia	263 (55)	17 (68)	246 (54.5)	0.188
TIA/CVA	82 (17)	8 (32)	74 (16)	0.043
Heart Failure *				0.004
HFrEF	16 (3.3)	1 (4)	15 (3.3)	
HFpEF	35 (7.3)	6 (24)	29 (6.4)	
Ischemic heart disease	119 (25)	6 (24)	113 (25)	0.910
Structural heart disease †	83 (17)	8 (32)	75 (17)	0.047
Atrial fibrillation	88 (18)	10 (40)	78 (17)	0.004
Beta blocker use	193 (40)	17 (68)	176 (39)	0.004
ND-CCB use	4 (0.8)	0	4 (0.9)	0.637
Antiarrhythmic drugs	18 (3.8)	0	18 (4)	0.310
COPD	40 (8.4)	5 (20)	35 (8)	0.031

Abbreviations: TIA, transient ischemic attack; CVA, cerebrovascular accident; HF, heart failure; rEF, reduced ejection fraction; pEF, preserved ejection fraction; ND-CCB, non-dihydropyridine calcium channel blockers; COPD, chronic obstructive pulmonary disease. *: Reduced ejection fraction includes left ventricular ejection fraction of 40% or less. †: Echocardiographic evidence of either hypertensive/ischemic/valvular heart disease or cardiomyopathy.

**Table 3 jcm-11-04781-t003:** Clinical variables and comparison between diagnostic and non-diagnostic Holter.

	Study Cohort *n* = 478 (%)	Diagnostic Holter *n* = 25 (%)	Non-Diagnostic *n* = 453 (%)	*p* Value
Time from event to Holter, mean ± SD, d	3.22 ± 3.23	2.04 ± 2.05	3.28 ± 3.35	<0.001
Time from admission to Holter, mean ± SD, d	2.63 ± 2.63	1.24 ± 0.92	2.71 ± 2.67	<0.001
Recurrent syncope *	82 (17)	6 (24)	76 (17)	0.351
Prior syncope admission	72 (15)	2 (8)	70 (15.5)	0.309
Pre-syncopal symptoms †	226 (47)	9 (36)	217 (48)	0.242
Chest pain	22 (4.6)	0	22 (5)	0.259
Palpitations	26 (5.4)	0	26 (6)	0.218
Event during effort	23 (4.8)	1 (4)	22 (5)	0.846
Event at rest	150 (31)	10 (40)	140 (40)	0.340
Oxygen saturation ≤ 93% §	39 (8.1)	1	38 (8)	0.435
Fever > 38 °C	13 (3.0)	0	13 (3)	0.390
Hemoglobin (g/dL) ¶	13.0 ± 1.8	12.8 ± 1.5	13.0 ± 1.8	0.584
Cardiac exam findings #	114 (24)	9 (36)	105 (23)	0.143
Abnormal ECG	245 (51)	18 (72)	227 (50)	0.033
Traumatic syncope	46 (9.6)	4 (16)	42 (9)	0.267
Hospital stay duration, mean ± SD, d	7.4 ± 6.4	9.6 ± 9.9	7.3 ± 6.1	0.260
In hospital death	2 (0.4)	0	2 (0.4)	0.739

* More than one syncopal event during the seven days before admission. †: Symptoms before syncopal event including dizziness, headache, nausea, and blurred vision. §: Oxygen saturation was measured at rest in ambient air. ¶: First result upon admission, presented as mean and standard deviation. #: Description of a murmur or an irregular rhythm.

**Table 4 jcm-11-04781-t004:** Linear regression analysis of risk factors for longer hospital stay duration.

Variable	Univariate Analysis		Multivariate Analysis	
	Standardized β	*p*	Standardized β	*p*
Age	0.15	<0.01	0.08	0.07
Heart disease *	0.06	0.17	0.01	0.78
COPD	−0.01	0.76	−0.03	0.51
Abnormal ECG	0.11	0.02	0.07	0.11
Time from admission to Holter	0.34	<0.01	0.33	<0.01

* Including patients with structural heart disease and/or heart failure.

**Table 5 jcm-11-04781-t005:** Multivariate analysis of risk factors for diagnostic Holter monitoring based upon univariate analysis.

Variable	Univariate Analysis		Multivariate Analysis	
	Odds Ratio (95% CI)	*p*	Odds Ratio (95% CI)	*p*
Age per 5 years	1.86 (1.61–2.15)	<0.01	1.35 (1.08–1.68)	<0.01
Reduced EF	1.52 (0.19–12.11)	0.69	1.31 (0.16–10.66)	0.80
Preserved EF	4.70 (1.73–10.11)	<0.01	4.10 (1.43–11.72)	<0.01
Event to Holter duration	0.78 (0.67–0.93)	<0.01	0.73 (0.56–0.96)	0.02

Abbreviations: CI, confidence interval; EF, ejection fraction.

**Table 6 jcm-11-04781-t006:** Multivariate analysis of risk factors for diagnostic Holter monitoring based upon clinically relevant factors *.

Variable	Univariate Analysis		Multivariate Analysis	
	Odds Ratio (95% CI)	*p*	Odds Ratio (95% CI)	*p*
Age	1.07 (1.02–1.11)	<0.01	1.05 (0.99–1.10)	0.06
Heart disease †	2.65 (1.15–6.10)	0.022	1.79 (0.75–4.27)	0.19
Beta blocker use	3.33 (1.41–7.88)	<0.01	2.36 (0.98–5.68)	0.06
Cardiac exam findings	1.86 (0.80–4.34)	0.14	1.21 (0.50–2.95)	0.67
Abnormal ECG	2.56 (1.05–6.25)	0.03	1.53 (0.59–3.96)	0.38
Traumatic syncope	1.86 (0.61–5.68)	0.27	1.77 (0.56–5.60)	0.33

Abbreviations: CI, confidence interval. *: Variables were chosen by the research team based on their clinical relevance. †: Including patients with structural heart disease and/or heart failure.

## Data Availability

Due to ethical and privacy concerns, the primary dataset cannot be made openly available. All other results and statistical analyses are within the manuscript. The study was conducted retrospectively, and according to the regulations of our institution review board, such data can be openly shared. Request for the dataset supporting our results can be made via helsinki@tlvmc.gov.il and will be given by the first author after approval.

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
