# Peer review of "Holter ECG for Syncope Evaluation in the Internal Medicine Department—Choosing the Right Patients"

_jcm, 2022, doi:10.3390/jcm11164781_

Round 1
Reviewer 1 Report
The work is well constructed and very interesting. I find the notion of overuse of Holter ECG in internal medicine department very relevant.
Different elements could be improved:
1. Why only 478 of the 1058 Holter ECGs were included? Other indication than syncope? Holter not completed or not available? I think it would be interesting to have this information, maybe in the form of a flow chart?
2. Only one patient with reduced EF in the Holter diagnostic group. Difficult to conclude its impact in a multivariate analysis. The structural heart disease variable includes 8 patients, whereas HFrEF and HFpEF added together give us 7 patients. Is this normal? Perhaps integrating the structural heart disease variable with or without reduction EF could be interesting in the multivariate model.
3. I am surprised that the following variables did not give interesting results in multivariate analysis = Abnormal ECG findings, traumatic syncope, betablocker use. These are important predictors of risk stratification for syncope (We often use a threshold of p-value < 0.2 to select the variables to be included in the multivariate analysis model. This condition may not be respected for clinically relevant variables such as traumatic syncope). It could be interesting to put on the same table the univariate analysis with OR and the multivariate analysis to illustrate all the variables tested and the variables selected to build the multivariate model.
4. Line 224 = I think that the cited studies are about Holter ECG made in ambulatory after a visit to the emergency and not in hospital monitoring. The framework of your study is the HolterECG in medicine department due to lack of bed with telemetry, so we are talking about patients with an indication of hospitalization for monitoring and not ambulatory patients. The question that can be asked is: What is the reason for hospitalization of this patients? Syncope or another?
Author Response
Reviewer #1:
The work is well constructed and very interesting. I find the notion of overuse of Holter ECG in internal medicine department very relevant.
Different elements could be improved:
- Why only 478 of the 1058 Holter ECGs were included? Other indication than syncope? Holter not completed or not available? I think it would be interesting to have this information, maybe in the form of a flow chart?
Thank you for highlighting this issue and for your valuable suggestion. Of the 1058 Holter tests, 561 were excluded as they were made for other indications than syncope. Additional 19 patients did not have available Holter recordings or access to their in-patient records. As recommended, we included a flow chart describing the inclusion and exclusion process (Figure 1, appears in lines 69-93).
- Only one patient with reduced EF in the Holter diagnostic group. Difficult to conclude its impact in a multivariate analysis. The structural heart disease variable includes 8 patients, whereas HFrEF and HFpEF added together give us 7 patients. Is this normal? Perhaps integrating the structural heart disease variable with or without reduction EF could be interesting in the multivariate model.
Thank you for your influential remarks regarding this topic. We agree that due to the low number of patients with HFrEF, especially those with a diagnostic Holter, the multivariate analysis is sub-optimal. Under structural heart disease we included patients with ventricular hypertrophy, significant valvular heart disease and ischemic cardiomyopathy – which resulted in the large number of patients compared with heart failure (some of the patients are included in both groups). Accordingly, in an additional multivariate analysis, we included a new variable of “heart disease” which includes patients with heart failure and/or structural heart disease (Table 6, appears in lines 213-217).
- I am surprised that the following variables did not give interesting results in multivariate analysis = Abnormal ECG findings, traumatic syncope, betablocker use. These are important predictors of risk stratification for syncope (We often use a threshold of p-value < 0.2 to select the variables to be included in the multivariate analysis model. This condition may not be respected for clinically relevant variables such as traumatic syncope). It could be interesting to put on the same table the univariate analysis with OR and the multivariate analysis to illustrate all the variables tested and the variables selected to build the multivariate model.
We agree with the reviewer about the mentioned important predictors for risk stratification. We therefore performed an additional multivariate analysis using the above predictors. To this model we also added additional important variables, after a consultation by the research team (Table 6, appears in lines 213-217). We would like to thank the reviewer for his suggestion, and accordingly, the odds ratio from both univariate and multivariate analysis appear in Tables 5 and 6. The above information was also added to the statistical analysis (lines 142-144).
- Line 224 = I think that the cited studies are about Holter ECG made in ambulatory after a visit to the emergency and not in hospital monitoring. The framework of your study is the HolterECG in medicine department due to lack of bed with telemetry, so we are talking about patients with an indication of hospitalization for monitoring and not ambulatory patients. The question that can be asked is: What is the reason for hospitalization of this patients? Syncope or another?
This is an important issue; however, we are not sure which studies the reviewer refers to (might be a change in line numbers between the manuscript versions). We agree that our main aim was to address the in-patient setting, where one of the reasons to order a Holter ECG is the lack of telemetry beds (other reasons might be the lack of skilled team or lack of staff that can perform continuous surveillance over the telemetry recordings). The vast majority of patients in our study were admitted due to syncope. Patients were only stratified by their reason for Holter and not their initial reason for admission. We believe that once a syncopal event occurs, regardless of the initial reason for admission, the question about the timing of Holter remains the same (in-patient vs. ambulatory) and hence our results can be generalized to hospitalized patients with syncope.
Reviewer 2 Report
Summary
Freund et al. reported in a retrospective study on the role of Holter ECG monitoring in patients admitted to an internal medicine department with a history of syncope. I found this article very interesting on the role of Holter ECG monitoring in this category of patients. However, some minor issues should be addressed.
Materials and Methods
· Page 2, line 62-64:
“Features of the syncopal episode included the episode setting (during activity, rest or after change in position) and symptoms before the event such as pre-syncope related symptoms (lightheadedness, nausea, feeling hot or cold, change in vision), chest pain or palpitations. “
This sentence is not so clear.
Results
· Page 5, line 153-154:
“We found a significant correlation between longer hospital stay and longer time from admission to Holter, r(479) =0.342, P<0.001.”
As reported by the authors in the discussions, this does not denote a definite causal role. A multivariate linear regression would have to be performed.
· Page 5, 167-171 line and Table 4:
“multivariate regression analysis (Table 4) revealed that HF with preserved ejection fraction (HFpEF, OR 4.1, 95% CI 1.43-11.72,P=0.008), older age per 5 years (OR 1.35, 95% CI 1.08-1.68, P=0.008) and a shorter duration from event to Holter initiation (OR 0.73, 95% CI 0.56-0.96, P=0.02) were independent predictors for a diagnostic Holter monitoring “
1)It would be necessary to report the results of the univariate logistic analysis in Table 4 in addition to the multivariate analysis. 2) Furthermore, were the characteristics of syncope included in the univariate analysis? Indeed, it is likely that the shorter time to Holter ECG monitoring in patients with diagnostic Holter may be due to clinical suspicion.
Author Response
Reviewer #2:
Freund et al. reported in a retrospective study on the role of Holter ECG monitoring in patients admitted to an internal medicine department with a history of syncope. I found this article very interesting on the role of Holter ECG monitoring in this category of patients. However, some minor issues should be addressed.
Materials and Methods
- Page 2, line 62-64:
“Features of the syncopal episode included the episode setting (during activity, rest or after change in position) and symptoms before the event such as pre-syncope related symptoms (lightheadedness, nausea, feeling hot or cold, change in vision), chest pain or palpitations. “
This sentence is not so clear.
We agree and would like to thank the reviewer for his observation. Accordingly, the sentence was changed.
Results
- Page 5, line 153-154:
“We found a significant correlation between longer hospital stay and longer time from admission to Holter, r(479) =0.342, P<0.001.”
As reported by the authors in the discussions, this does not denote a definite causal role. A multivariate linear regression would have to be performed.
Thank you for this valuable suggestion. We agree that a multivariate linear regression is appropriate and accordingly it was added to the manuscript (Table 4, lines 194-195). It was also mentioned in the discussion (line 259).
- Page 5, 167-171 line and Table 4:
“multivariate regression analysis (Table 4) revealed that HF with preserved ejection fraction (HFpEF, OR 4.1, 95% CI 1.43-11.72,P=0.008), older age per 5 years (OR 1.35, 95% CI 1.08-1.68, P=0.008) and a shorter duration from event to Holter initiation (OR 0.73, 95% CI 0.56-0.96, P=0.02) were independent predictors for a diagnostic Holter monitoring “
1)It would be necessary to report the results of the univariate logistic analysis in Table 4 in addition to the multivariate analysis.
Thank you for highlighting this issue. We changed the mentioned Table to also include the odds ratio of each variable in the univariate analysis (appear in Table 5, lines 208-211).
2) Furthermore, were the characteristics of syncope included in the univariate analysis? Indeed, it is likely that the shorter time to Holter ECG monitoring in patients with diagnostic Holter may be due to clinical suspicion.
Syncope characteristics, including the setting, symptoms before the event and any prior events were included in the univariate analysis (Table 3, lines 168-169). These characteristics were similar between the groups.
We agree with the reviewer that clinical suspicion might explain the relationship between shorter time to Holter and a diagnostic Holter. However, as stated in the syncope guidelines and seen also among patients with ischemic stroke, early monitoring is preferred and has a higher yield in detecting an underlying arrhythmia. To fully address this issue, prospective studies with a randomized control group are needed.
Round 2
Reviewer 1 Report
I thank the authors of modifications that significantly improve the article.
1. Why did you make 2 different tables concerning the diganostic perdicator at the Holter ECG? Could integration of all the variables not bring different results? Or at least the significant variables in univariate (betablockers, abnormal ECG findings).
2. "We believe that once a syncopal event occurs, regardless of the initial reason for admission, the question about the timing of Holter remains the same (in-patient vs. ambulatory) and hence our results can be generalized to hospitalized patients with syncope."
My point is that in the management of syncope it is necessary to stratify the risk of rhythmic disorders. As indicated by the ESC guidelines, following the initial clinical evaluation and depending on the risk stratification, the patient will be hospitalized or managed as an outpatient.
Hospitalized patient = high risk = Telemetry
Ambulatory patient = Low risk = Holter if frequency of symptoms ≥ 1 per week.
In the discussion when reference 5 was mentioned, I find the explanation confusing. In your work we talk about the place of holter in high risk patients only.
Author Response
Reviewer #2:
- Why did you make 2 different tables concerning the diganostic perdicator at the Holter ECG? Could integration of all the variables not bring different results? Or at least the significant variables in univariate (betablockers, abnormal ECG findings).
We would like to thank you for your interest and in-depth review. We agree that an explanation is missing in this case. Our initial multivariate analysis (currently in Table 5) was done using backward stepwise regression among the significant variables in the univariate analysis. This method chooses the best model by removing less significant factors. After the first revision, we have added an additional multivariate regression (Table 6) of significant factors in the univariate analysis (including 2 additional clinically relevant factors). The second regression use the enter method, which includes all variables regardless of their effect on the model. In accordance, we included this explanation in the statistical analysis section (lines 141 and 144).
- "We believe that once a syncopal event occurs, regardless of the initial reason for admission, the question about the timing of Holter remains the same (in-patient vs. ambulatory) and hence our results can be generalized to hospitalized patients with syncope."
My point is that in the management of syncope it is necessary to stratify the risk of rhythmic disorders. As indicated by the ESC guidelines, following the initial clinical evaluation and depending on the risk stratification, the patient will be hospitalized or managed as an outpatient.
Hospitalized patient = high risk = Telemetry
Ambulatory patient = Low risk = Holter if frequency of symptoms ≥ 1 per week.
In the discussion when reference 5 was mentioned, I find the explanation confusing. In your work we talk about the place of holter in high risk patients only.
Thank you for highlighting this issue. We agree that only patients with high-risk for an arrhythmia related syncope should have in-hospital monitoring. As you mentioned, our work addressed high risk patients, which were hospitalized and underwent inpatient Holter monitoring. Our aim was to test if Holter ECG has a place in the hospital setting for these patients. In the mentioned paragraph, we wanted to address the possible overuse of ECG monitoring among patients with syncope. Following your important remark, we changed the paragraph to be more specific about how Holter is regarded by the ESC guidelines (lines 271-276) and we hope our message is now better understood.